# Occurrence and Management of Immunotherapy-Associated Adverse Events in Patients with Gynecological Cancers

**DOI:** 10.3390/cancers16071371

**Published:** 2024-03-30

**Authors:** Ina Shehaj, Maria Schröder, Valerie Catherine Linz, Slavomir Krajnak, Katrin Almstedt, Kathrin Stewen, Roxana Schwab, Annette Hasenburg, Marcus Schmidt, Anne-Sophie Heimes

**Affiliations:** Department of Gynecology and Obstetrics, University Medical Center, Johannes Gutenberg-University Mainz, 55131 Mainz, Germany; ina.shehaj@unimedizin-mainz.de (I.S.); maria.schroeder@unimedizin-mainz.de (M.S.); valerie.linz@unimedizin-mainz.de (V.C.L.); slavomir.krajnak@unimedizin-mainz.de (S.K.); katrin.almstedt@unimedizin-mainz.de (K.A.); kathrin.stewen@unimedizin-mainz.de (K.S.); roxana.schwab@unimedizin-mainz.de (R.S.); annette.hasenburg@unimedizin-mainz.de (A.H.); marcus.schmidt@unimedizin-mainz.de (M.S.)

**Keywords:** immunotherapy-related adverse events, immune checkpoint inhibitors, gynecological malignancies

## Abstract

**Simple Summary:**

The implementation of immune checkpoint inhibitors into the therapeutic armamentarium for many solid tumors has transformed the treatment landscape of gynecological malignancies. The mechanism of action of immune checkpoint inhibitors is to increase the body’s own tumor-directed T-cell response, which can, however, lead to a new spectrum of immunotherapy-associated adverse events (irAEs). In the present study, we retrospectively analyzed the incidence, diagnosis, and management of irAEs in patients with gynecologic malignancies who received immune checkpoint inhibitors and discussed our findings in the context of the recent literature. Our results emphasize the need for proactive monitoring and tailored management strategies to optimize the safety and efficacy of immunotherapy in cancer patients.

**Abstract:**

Background: Immune checkpoint inhibitors (ICIs) have emerged as an essential therapeutic approach in treating many solid tumors. ICIs enhance the body’s anti-tumor T-cell activity, resulting in a novel spectrum of immunotherapy-related side effects. This novel spectrum of adverse events differs significantly from the side effects of conventional chemotherapy. It, therefore, requires special attention in the diagnosis and management of immunotherapy-related adverse events (irAEs). The present study aimed to retrospectively analyze the incidence, diagnosis, and management of irAEs in patients with gynecologic malignancies who received ICIs and to discuss these findings in the context of the recent literature. Methods: In the present retrospective overview, we evaluated patients with gynecologic malignancies (breast, endometrial, cervical, ovarian) who received ICIs with regard to the incidence, type, and time to onset of irAEs. A total of 61 patients treated at the Department of Gynecology and Obstetrics, University Medical Center Mainz, Germany, between 2018 and 2023 were included in the analysis. Results: A total of 32.8% of patients developed an irAE of any grade or type. The median time to irAE was 24 weeks. The most frequently observed irAEs were grade 1 (20%) or 2 (35%). Immunotherapy-related grade 3 or 4 adverse events occurred in 45% of patients (40% grade 3, 5% grade 4). The most common type of irAE in our cohort was hypothyroidism, followed by hepatitis and colitis. Cox regression analysis identified the duration of ICI therapy as the only significant factor influencing the incidence of irAEs (*p* = 0.004). Conclusion: The broad spectrum of irAEs and the onset time of irAEs are important challenges of therapy with ICIs, requiring proactive monitoring and tailored management strategies to optimize the safety and efficacy of immunotherapy.

## 1. Introduction

Implementing immune checkpoint inhibitors (ICIs) into the therapeutic armamentarium for many solid tumors represents a major advance in the search for more effective and targeted cancer therapies [1]. Since 1957, when Thomas and Burnett suggested that tumor cells could induce an immune response, a number of immunotherapeutic strategies have been approved by the Food and Drug Administration (FDA). Cytotoxic T-lymphocyte-associated Protein 4 (CTLA-4) was the first major immunological checkpoint. Preclinical experiments in 1996 showed that CTLA-4 blockade with monoclonal antibody therapy resulted in tumor regression in mice [2]. In 2011, the FDA approved the first antibody to block CTLA-4, ipilimumab, for the treatment of advanced melanoma. In the decade following this approval, several ICIs that block other immunological checkpoint pathways such as programmed cell death 1 (PD-1) and its ligand, PD-L1, have been approved by the FDA. Although melanoma is considered one of the most immunogenic malignancies, the therapeutic indications for ICIs are increasing in other tumors [3]. 

Immune checkpoint inhibitors in combination with chemotherapy are also an integral part of guideline-based therapy for gynecological tumor entities such as early triple-negative breast cancer [4,5], advanced triple-negative breast cancer [6,7], metastatic endometrial cancer [8,9], and metastatic cervical cancer [10,11]. Table 1 provides an overview of the phase III evidence that includes an immune checkpoint inhibitor for treating gynecologic malignancies.

ICIs, with their ability to block inhibitory pathways such as anti-programmed death-ligand 1 (PD-1/PD-L1) and cytotoxic T-lymphocyte-associated Protein 4 (CTLA-4), have emerged as a cornerstone of cancer immunotherapy. By releasing the brakes on immune surveillance, these agents enable the immune system to recognize and eliminate cancer cells more effectively [13]. However, the increased immune activity associated with this therapeutic approach also leads to a spectrum of immune-related adverse events (irAE), requiring careful diagnosis and management to ensure safety and efficacy [14].

The most common irAEs are those affecting the skin, often presenting as rash, pruritus, or dermatitis [15]. Gastrointestinal irAEs such as colitis and diarrhea are also common and can vary in severity. A notable immune-related adverse event that requires focused attention is immunotherapy-associated hepatitis [16]. Hepatitis, characterized by liver inflammation, is emerging as a notable issue in the context of immunotherapy. Elevated liver enzymes, icterus, and hepatomegaly may indicate the onset of immunotherapy-associated hepatitis, which requires prompt diagnosis and treatment. As hepatotoxicity is a potentially serious complication, vigilance and proactive monitoring of liver function are essential components of the comprehensive care framework for gynecologic cancer patients undergoing immunotherapy. Endocrine irAEs, including thyroid dysfunction and hypophysitis, are additional considerations given their potential impact on hormonal balance. Although less common, pneumonitis is a critical respiratory irAE that requires prompt attention. In addition, immune-related adverse events are not limited to specific organ systems, as they can manifest in various forms, including fatigue, arthralgia, and myalgia. These events, resulting from the activation of the immune system, underscore the delicate balance required to harness its therapeutic potential while mitigating unintended consequences.

Immunotherapy-related side effects usually occur in close temporal relation to the use of immune checkpoint inhibitors but can also occur months and years after the completion of therapy. It is, therefore, important to raise awareness of immunotherapy-related side effects among both patients and physicians. A review article by Martins et al. graphically compares the kinetics of the most important immunotherapy-related adverse events during immunotherapy with anti-PD(L)1 antibodies vs. anti-CTLA-4 antibody therapy and illustrates that immunotherapy-related adverse events differ significantly in their kinetics [17]. While skin reactions usually occur in close time relation to the immunotherapy, endocrinopathies can also occur with a time delay or persist even after the end of therapy. The kinetics of immunotherapy-associated side effects depend not only on the organ affected or the type of side effect, but also on the type of immune checkpoint inhibitor used, with clear differences between anti-CTLA-4 antibodies and anti-PD-(L)1 antibodies. 

Many oncology societies, such as the European Society for Medical Oncology (ESMO) or the American Society of Clinical Oncology (ASCO), have published recommendations for the diagnosis and management of immunotherapy-related AEs [14,15,18]. These recommendations also grade the severity of adverse events according to the Common Terminology Criteria for Adverse Events (CTCAE) version 5.0. on a scale from 1 to 5 (1 = mild, 2 = moderate, 3 = severe, 4 = life threatening, and 5 = death related to toxicity) and are, therefore, an important tool in daily clinical practice.

In the following study, we retrospectively analyzed the incidence, diagnosis, and management of immune-related adverse events in all patients with gynecological tumors at our clinic who received immune checkpoint inhibitor therapy and discussed our findings in the context of recent publications.

## 2. Materials and Methods

### 2.1. Study Population

A retrospective analysis of patients with gynecological cancers who received ICIs from February 2018 to June 2023 at the University Medical Center Mainz was conducted. The intervention included patients with locally advanced or early-stage triple-negative breast cancer (TNBC) at high risk of recurrence, locally recurrent unresectable or metastatic TNBC (PD-L1 with a CPS ≥ 10), advanced or recurrent endometrial carcinoma, recurrent or metastatic cervical cancer (PD-L1 with a CPS ≥ 1), and advanced ovarian cancer in a clinical trial setting. The frequency (in absolute and relative numbers) of the different tumor entities within the study cohort is shown in Figure 1. 

The analysis included drug regimens that used anti–PD-1 antibodies (pembrolizumab, dostarlimab, durvalumab) or the anti-PD-L1 antibody atezolizumab as monotherapy or in combination with chemotherapy, targeted therapy, or vaccines. 

The therapy was administered according to the multidisciplinary tumor board recommendation, and informed consent was obtained from all patients before treatment. 

We collected clinicopathological data, including characteristics of participants such as median age, ECOG (Eastern Cooperative Oncology Group) performance status, concomitant pathologies, treatment methods (monotherapy or combination targeted therapy), ICI agents, and irAEs data. We thoroughly evaluated the study participants’ electronic medical records. We extracted the following data: time to onset of irAE, nature and CTCAE grade of the irAE, date of rechallenge therapy with ICI, hospitalization and mortality associated with the irAE, and follow-up data.

Patients with pre-existent autoimmune diseases, hyperthyroidism, or cardiorespiratory dysfunction were excluded. 

### 2.2. Defining Immune-Related Adverse Events

In our study, the irAEs were classified according to the affected organ system. The severity of adverse events is graded according to the Common Terminology Criteria for Adverse Events (CTCAE) version 5.0. on a scale from 1 to 5 (1 = mild, 2 = moderate, 3 = severe, 4 = life threatening, and 5 = death related to toxicity) [14,18].

### 2.3. Outcomes

The primary outcome was the rate of occurrence of irAEs of any grade or type. Secondary endpoints were the time of onset of irAEs and their management.

### 2.4. Statistical Methods

Statistical analyses were performed using SPSS statistical software system version 27.0 (SPSS Inc., Chicago, IL, USA). A two-sided *p*-value < 0.05 was considered statistically significant. The descriptive analyses of patients’ characteristics were conducted using median and range for continuous data and relative and absolute frequencies for categorical data. The chi-square test was used for categorical data, and *t*-tests were used for normally distributed continuous data. Univariate Cox regression analysis was conducted to identify the role of several variables in the occurrence of irAEs. Using logistic regression for binary outcomes, we estimated 95% confidence intervals (CIs). Figure 2 was created with BioRender.com URL (accessed on 29 February 2024). 

## 3. Results

A total of 61 patients with gynecological malignancies treated with an ICI-containing regimen were enrolled. The median age was 57 years and ranged from 31 to 86 years. Of these patients, 55.5% had triple-negative breast cancer; a total of 31.1% received an ICI in combination with chemotherapy as first-line treatment in the metastatic setting, and 24.6% received an ICI in combination with chemotherapy in the neoadjuvant/post-neoadjuvant setting. A total of 42.6% of patients suffered from metastatic endometrial cancer (31.1%), metastatic cervical cancer (16.4%), or advanced/recurrent ovarian cancer (18%). Most patients presented with good ECOG performance status and 80.3% were categorized as ECOG 0 (59%) or ECOG 1 (21.3%). However, two patients were classified as ECOG 4. These patients were diagnosed with metastatic breast cancer or metastatic endometrial cancer and had received extensive prior treatment. In both cases, the decision to start immunotherapy was made using shared decision-making principles, taking into account the patient’s strong therapeutic preferences. The most common comorbidity was arterial hypertension (24.6%), followed by diabetes mellitus (9.8%), and bronchial asthma (4.9%). The most commonly used ICI in the study population was pembrolizumab (49.2%) in combination with chemotherapy or as maintenance therapy alone. Atezolizumab was used in 37.7% of cases, and 4.9% and 8.2% of patients were treated with durvalumab and dostarlimab, respectively. The duration of therapy (ICI as combination or monotherapy) varied from 4 to 148 weeks, with a median of 37 weeks. A total of 32.8% of patients developed any grade of irAE. The median time to onset of an irAE was 24 weeks. The most commonly observed irAEs were grade 1 (20%) or 2 (35%). Grade 3 or 4 immunotherapy-related adverse events occurred in 45% of patients (40% grade 3, 5% grade 4). Key patient characteristics and frequencies of immunotherapy-related adverse events are summarized in Table 2.

The most common irAE in the observed population was hypothyroidism, which in most cases (67%) could be successfully treated with L-thyroxine supplementation without the need to discontinue ICI therapy. Other common irAEs were hepatitis and colitis, both of which required discontinuation of ICI therapy and treatment with corticosteroids. Pneumonitis occurred as an immunotherapy-related adverse event in two patients, one of them receiving pembrolizumab and one with atezolizumab. In one of the two cases, maintenance pembrolizumab (in the post-neoadjuvant setting) had to be discontinued prior to completion. All types of irAEs are summarized with their frequencies in Table 3 and Figure 2.

Meanwhile, the management of immunotherapy-related adverse events with their corresponding frequencies is summarized in Table 4.

In the univariate Cox regression analysis, age, ECOG, and concomitant diagnosis showed no significant effect on the occurrence of irAEs (*p* > 0.05). Interestingly, combination therapy such as chemotherapy or targeted therapies did not play a significant role in the occurrence of irAEs (*p* = 0.227). The duration of ICI therapy was the only significant factor influencing the incidence of irAEs (*p* = 0.004). The duration of immunotherapy was defined as the time from initiation to completion/discontinuation of therapy (due to tumor progression or toxicity) and was a median of 32 weeks. It varied from 4 weeks as the shortest duration of therapy to 148 weeks as the longest duration of therapy (Table 5).

## 4. Discussion

PD(L)-1 antibodies are now established as an integral part of systemic therapy for many solid tumors, including gynecological malignancies such as breast cancer (early TNBC or metastatic TNBC), advanced endometrial cancer, advanced ovarian cancer, or metastatic cervical cancer. The adverse events profile can be explained by the mechanism of action and differs significantly from the side effect profile of standard chemotherapy. To ensure the effectiveness and quality of life during immunotherapy, both patients and physicians must be aware of immunotherapy-associated side effects. In our retrospective study, we evaluated the occurrence, type, and management of immunotherapy-associated side effects in a cohort of 61 patients with gynecological malignancies. Overall, therapy with immune checkpoint inhibitors was well tolerated. In general, the side effect profile of antibodies that inhibit PD-1 or PD-L1 differs from the adverse events of CTLA-4 antibodies, with anti-PD-(L)1 antibodies tending to be better tolerated [17]. 

The overall rate of immunotherapy-associated adverse events was approximately 33% in our collective of 61 patients with a gynecological malignancy and immunotherapy with an anti-PD-(L)1 antibody. This is in line with the rates of immunotherapy-associated adverse events reported in the literature. A retrospective analysis of the MD Anderson Cancer Centre reported that 34% of 290 patients with advanced solid tumors treated with immunotherapy developed any grade of irAE [19]. 

### 4.1. Immunotherapy-Associated Endocrinopathies: Hypothyroidism and Hyperthyroidism

The most common immunotherapy-associated adverse event in our cohort was hypothyroidism: six patients (9.9%) out of a total of 61 patients developed hypothyroidism. Of these, four were treated with pembrolizumab and two patients with atezolizumab. Our results are in line with the available literature, where hypothyroidism was described with a frequency of 6–9% as an immune-mediated side effect under therapy with anti-PD-1 or anti-PD-L1 antibodies [17]. In the majority of cases of hypothyroidism, immunotherapy could be continued with supplementation of L-thyroxine and close monitoring of laboratory chemistry in our patient collective, which is based on the recommendations and guidelines for the management of immunotherapy-associated side effects [14,18]. Hypothyroidism was also described in the literature as a very common endocrinopathy in the sense of an immunotherapy-associated side effect of PD-1 or PD-L1 antibody therapies [17,20]. Hypothyroidism occurred significantly more frequently in patients receiving PD-1/PD-L1 antibody therapy compared to patients receiving CTLA4 antibodies [17]. Typically, hypothyroidism as an immunotherapy-associated side effect can occur very early (after only six weeks) under immune checkpoint blockade [16]. In most cases, immunotherapy-associated hypothyroidism is asymptomatic and is diagnosed based on the corresponding laboratory constellation with increased TSH and decreased T3/T4.

Compared to hypothyroidism, hyperthyroidism is an immunotherapy-associated endocrinopathy, which is described much less frequently in the literature [18]. It can occur as transient thyroiditis, is clinically less symptomatic in more than half of cases, and is often followed by hypothyroidism. In our cohort, hyperthyroidism occurred only in one case (under durvalumab therapy), which was successfully treated with thyrostatic therapy and temporary interruption of durvalumab therapy in accordance with the guidelines.

### 4.2. Immunotherapy-Associated Hepatitis

Another common immunotherapy-associated side effect in our group was hepatitis.

For optimal management and to minimize immunotherapy-associated hepatitis, it is critical to evaluate liver function parameters (GOT, GPT, GGT, AP, bilirubin) and hepatic synthesis parameters (albumin, INR) by laboratory testing prior to initiating immune checkpoint inhibitor therapy and to rule out pre-existing liver disease and viral hepatitis. Patients with ICI-induced hepatitis most commonly present with isolated elevations of liver transaminases [17]. The reported incidence of immunotherapy-associated hepatitis in the literature varies according to the different agents: 2–15% in CTLA-4 inhibitors, 0–3% in PD-1 inhibitors, and 0–6% in PD-L1 inhibitors [21]. Our results are consistent with the published literature, as a total of four patients (6.6%) developed hepatitis after receiving pembrolizumab. It is important to note that the CTCAE grading of ICI-induced hepatitis is crucial, as the severity of the adverse events corresponds to the treatment. Three of our patients had grade 3 hepatitis, and one patient presented with grade 2 ICI-associated hepatitis. According to the ASCO guidelines, in all patients with grade 3 hepatitis, the therapy with ICI was discontinued, and therapy with corticosteroids was started. In the patient presenting with grade 2 hepatitis, pembrolizumab immunotherapy was temporarily suspended, and corticosteroids were initiated.

### 4.3. Immunotherapy-Associated Colitis

Another common immunotherapy-associated side effect is colitis, which is described in 1–10% of cases of anti-PD(L1) antibody therapy in the literature [17] and manifests clinically with diarrhea. After thyroid dysfunction and hepatitis, colitis was the third most common immunotherapy-associated adverse event in our cohort. A total of 3/61 (5%) patients developed colitis, with one case each under pembrolizumab, atezolizumab, and dostarlimab, respectively. In all cases, the immunotherapy-associated colitis manifested with diarrhea; in one case, inpatient treatment with symptomatic therapy and corticosteroid therapy was required, as well as temporary discontinuation of immune checkpoint inhibitor therapy. In one of the three cases of immunotherapy-associated colitis, a synchronous infection with Clostridium difficile was diagnosed in cultures, which was treated with vancomycin. The guideline-compliant treatment of immunotherapy-associated colitis is shown in Table 6. 

### 4.4. Immunotherapy-Associated Pneumonitis

ICI-induced pneumonitis, defined as lung tissue inflammation in patients receiving ICI is a relatively common adverse event. Most patients present with persistent cough, chest pain, or dyspnea, but some of them are asymptomatic, and the diagnosis will be made in the routine CT scan controls under therapy. In a systematic review, pneumonitis occurred in 4% of patients receiving PD-(L)1 inhibitors (1% grade ≥ 3) and in 1% of patients receiving anti-CTLA-4 therapy (1% grade ≥ 3) [22]. The results of the above-mentioned study were similar to our analysis, in which two patients (3.3%) presented with ICI-associated pneumonitis. The patients received pembrolizumab and atezolizumab, respectively. Both patients in our cohort presenting with pneumonitis Grade 2 were treated with corticosteroids, and ICIs were withheld permanently in accordance with current clinical practice guidelines. 

Interestingly, in the literature, a decreased prevalence of ICI-induced pneumonitis could be shown in patients with advanced disease receiving ICI in the second-line setting. These results could be explained by the fact that patients in this setting have suppressed or compromised immune system function [17]. 

### 4.5. Immunotherapy-Associated Neurological Side Events

The clinical spectrum of neurologic irAEs is heterogeneous, as they can affect the central or peripheral nervous system. Neurotoxicity is a rare adverse event associated with ICIs. Larkin et al. published a retrospective cohort study in a population of more than 3700 patients with advanced melanoma treated with ICIs, showing an incidence of neurologic irAEs of approximately 1% [23]. In our analysis, we also observed only one patient (1.6%) with atezolizumab who experienced a neurologic irAE, which was defined as grade 3 vertigo. As expected, immunotherapy-induced neurologic adverse events were more common in patients receiving chemotherapy. In a meta-analysis of 39 studies, Farooq et al. evaluated the risk of neurological adverse events in patients treated with ICI. Three subgroups were identified for analysis: the ICI alone subgroup, the ICI and chemotherapy subgroup, and the placebo subgroup. Interestingly, the results showed that the use of ICI as monotherapy was associated with an increased risk of irAEs compared to placebo (HR 1.57, 95% CI: 1.30–1.89) and that the risk was much lower compared to chemotherapy in combination with ICI (HR 0.22, 95% CI: 0.13–0.39) [24]. 

The heterogeneity of therapies, as well as the heterogeneity of the observed study cohort, may limit the interpretation of our analysis, as some patients received combination therapy. Interestingly, we showed that the combination therapy did not significantly impact the occurrence of irAEs. Furthermore, treatment duration was the only independent variable for the occurrence of immunotherapy-related adverse events.

Further limitations of our analysis were the small number of cases, the short follow-up period, and thus the lack of possibility of correlating the immunotherapy-associated side effects with the clinical outcome. In a retrospective analysis by Fujii et al., the occurrence and severity of the immunotherapy-associated side effects were associated with a better response rate [19]. Another potential weakness of our evaluation was the retrospective design.

ICIs are being used in an ever-increasing number of patients and are increasingly being used with curative intent. IrAEs represent a differential diagnostic and therapeutic challenge in daily clinical practice. In our collective, the treatment to manage irAEs was successful. Table 7 summarizes general strategies for managing irAEs in the setting of immune checkpoint inhibitor treatment, taking into account current guidelines. In steroid-refractory irAE, biopsy should be considered, and treatment should be managed by a multidisciplinary team, including (targeted) immunomodulatory therapy if necessary [25].

## 5. Conclusions

In conclusion, we described a single-center analysis of immune-related adverse events in patients with gynecologic malignancies. The broad spectrum of irAEs, potentially affecting any organ, and the time of onset of irAEs remain the two most important points of our discussion, considering their important impact on the management of irAEs. Therefore, clinicians should perform a thorough evaluation of patients receiving ICIs to minimize the risk of irAEs not being detected in time, which could affect the oncological therapy and outcome of these patients.

## Figures and Tables

**Figure 1 cancers-16-01371-f001:**
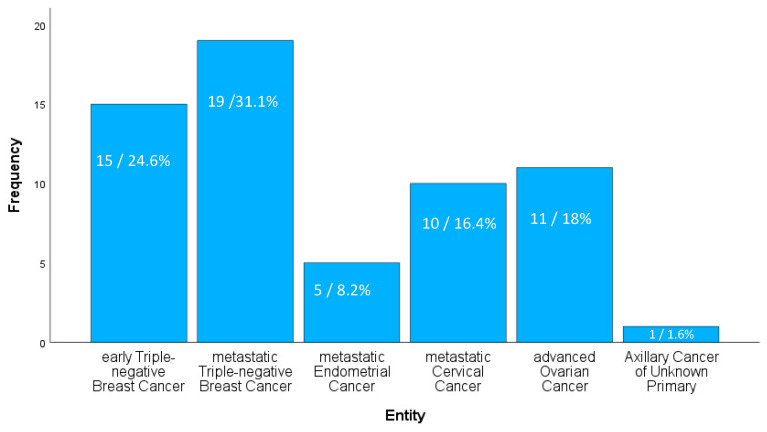
Frequency (in absolute and relative numbers) of different entities in the retrospective study cohort.

**Figure 2 cancers-16-01371-f002:**
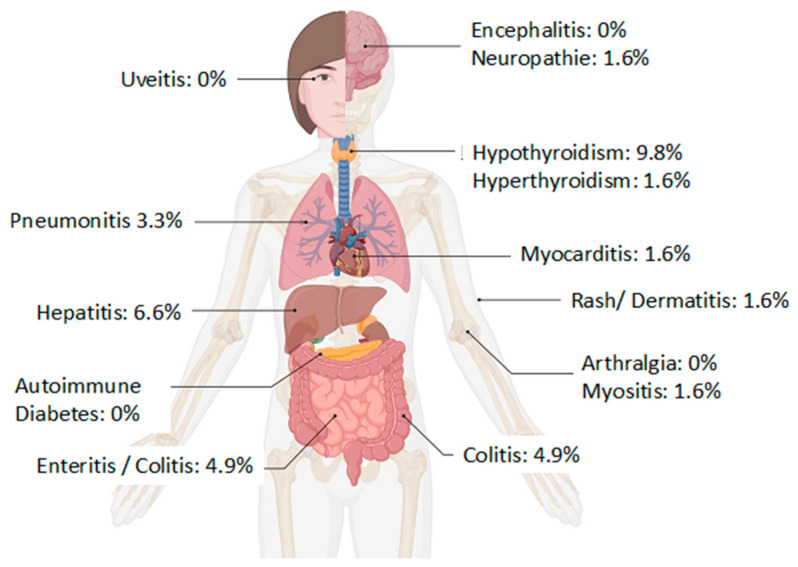
Frequency of irAEs in the study cohort of 61 patients.

**Table 1 cancers-16-01371-t001:** Randomized evidence for ICIs in patients with gynecological malignancies.

Entity	Author/Study	Therapy	Efficacy	irAEs of Any Grade and Any Type
Early triple-negative breast cancer	Schmid et al. KEYNOTE 522 [4]	Chemotherapy +/− pembrolizumab	pCR ITT: 64.8% vs. 51.2%	25% vs. 5.8%
Metastatic triple-negative breast cancer	Cortes et al. KEYNOTE 355 [6]	Chemotherapy +/− pembrolizumab	PFS ITT: 7.5 months vs. 5.6 months HR 0.82 (0.69–0.97)	26% vs. 6%
Metastatic triple-negative breast cancer	Schmid et al. IMPASSION 130 [7]	Nab-paclitaxel +/− Atezolizumab	PFS (months) ITT: 7.2 vs. 5.5 HR 0.80 (0.69–0.92)	24.8% vs. 8.4%
Advanced/recurrent/metastatic/endometrial cancer	Mirza et al. RUBY [8]	Carboplatin/Paclitaxel +/− Dostarlimab	PFS dMMR-cohort (after 24 months):61.4% vs. 15.7%; HR: 0.28 (95%-CI: 0.6–0.50); *p* < 0.001 PFS ITT (after 24 months): 36.1% vs. 18.1%; HR: 0.64 (95%-CI: 0.51–0.80); *p* < 0.001.	29.4% vs. 12.1%
Advanced/recurrent/metastatic/endometrial cancer	Eskander et al. NRG GY018 [12]	Carboplatin/Paclitaxel +/− Pembrolizumab	PFS (after 12 months): dMMR-cohort: 74% vs. 38%; HR: 0.30 (95%-CI: 0.19–0.48); *p* < 0.001	38.5% vs. 26.4%
			PFS (months) pMMR-cohort:13.1 vs. 8.7 HR: 0.54 (95% CI: 0.41–0.71); *p* < 0.001	33.3% vs. 19.7%
Metastatic/Advanced endometrial cancer (2nd line)	Makker et al. KEYNOTE 775 [9]	Lenvatinib + Pembrolizumab vs. chemotherapy (physician’s choice)	PFS (months): pMMR population: 6.6 vs. 3.8 HR 0.60; 95% CI: 0.50 to 0.72; *p* < 0.001; ITT: 7.2 vs. 3.8 months; HR 0.56; 95% CI, 0.47 to 0.66; *p* < 0.001	57.4% vs. 0.8%
Metastatic cervical cancer	Colombo et al. KEYNOTE 826 [10,11]	Carboplatin/Paclitaxel/Bevacizumab +/− Pembrolizumab	PFS (months) ITT: 10.4 vs. 8.2 HR 0.65; 95% CI, 0.53−0.79; *p* < 0.001	33.9% vs. 15.2%

Abbreviations: irAE—immunotherapy-related adverse events; pCR—pathological complete response; ITT—intention-to-treat; PFS—progression-free survival; dMMR—deficient mismatch repair; pMMR—proficient mismatch repair.

**Table 2 cancers-16-01371-t002:** Patients’ characteristics.

Variables	Numbers	Percentage
Age in years (median, range)	57 (31–86)	-
ECOG		
0	36	59
1	13	21.3
2	8	13.1
3	2	3.3
4	2	3.3
Concomitant diagnosis		
Arterial hypertension	15	24.6
Asthma	3	4.9
Gastritis	1	1.6
Glaucoma	1	1.6
Chronic renal insufficiency	1	1.6
Metabolic syndrome	1	1.6
Nervus opticus atrophy	1	1.6
Diabetes mellitus	6	9.8
Entity		
Early triple-negative breast cancer	15	24.6
Metastatic triple-negative breast cancer	19	31.1
Metastatic endometrial cancer	5	8.2
Metastatic cervical cancer	10	16.4
Advanced ovarian cancer	11	18.0
Adeno CUP axilla	1	1.6
Immune Checkpoint Inhibitor		
Pembrolizumab	30	49.2
Atezolizumab	23	37.7
Dostarlimab	5	8.2
Durvalumab	3	4.9
Duration of Therapy in weeks (median, range)	32 (4–148)	-
Immune-related adverse events		
Yes	20	32.8
No	41	67.2
Time to onset of irAEs in weeks (median)	24	
Common Terminology Criteria for Adverse Events		
Grade 1	4	20
Grade 2	7	35
Grade 3	8	40
Grade 4	1	5

**Table 3 cancers-16-01371-t003:** Summary of immune-related adverse events associated with an immune checkpoint inhibitor.

Adverse Event (Nr., %)	Overall Toxicity (N = 61)	Pembrolizumab (N = 30)	Atezolizumab (N = 23)	Durvalumab (N = 3)	Dostarlimab (N = 5)
Colitis	3 (4.9)	1 (3.3)	1 (4.3)	0 (0)	1 (20)
Pneumonitis	2 (3.3)	1 (3.3)	1 (4.3)	0 (0)	0 (0)
Hepatitis	4 (6.6)	4 (13.3)	0 (0)	0 (0)	0 (0)
Stomatitis	1 (1.6)	1 (3.3)	0 (0)	0 (0)	0 (0)
Hypothyroidism	6 (9.8)	4 (13.3)	2 (8.7)	0 (0)	0 (0)
Hyperthyroidism	1 (1.6)	0 (0)	0 (0)	1 (33.3)	0 (0)
Myokarditis	1 (1.6)	1 (3.3)	0 (0)	0 (0)	0 (0)
Myositis	1 (1.6)	1 (3.3)	0 (0)	0 (0)	0 (0)
Neuropathie	1 (1.6)	0 (0)	1 (4.3)	0 (0)	0 (0)
Skin rash	1 (1.6)	0 (0)	1 (4.3)	0 (0)	0 (0)
Nephritis	1 (1.6)	0 (0)	0 (0)	1 (33.3)	0 (0)

**Table 4 cancers-16-01371-t004:** Management of immune-related adverse events.

Variables	Numbers	Percentage
Management/Protocol		
Hold ICPi Therapy, initiate corticosteroids	14	70
Continue ICPi, close monitoring	1	5
Continue ICPi, begin with L-Thyroxin	4	20
Hold ICPi, begin with methimazole	1	5
Hospitalization		
Yes	3	15
No	17	85

Abbreviations: ICI—immune checkpoint inhibitors.

**Table 5 cancers-16-01371-t005:** Factors Associated with the occurrence of Immune-Related Adverse Event.

Variables	Univariate Analysis OR (95% CI)	*p*-Value
Age	0.99 (0.96–1.03)	0.651
ECOG	0.99 (0.52–1.89)	0.985
Concomitant diagnosis	0.91 (0.77–1.07)	0.239
Combination therapy	0.66 (0.33–1.23)	0.227
Duration of therapy	0.91 (0.86–0.97)	0.004

Abbreviations: OR—odds ratio; CI—confidence interval; ECOG—Eastern Cooperative Oncology Group.

**Table 6 cancers-16-01371-t006:** Approach and management: immunotherapy-associated colitis.

CT CAE Grade	Management
Grade 1	continue ICI, symptomatic therapy, stool cultures for pathogenesis germs
Grade 2	discontinue ICI, symptomatic therapy, 1 mg/kgKG prednisone equivalent
Grade 3	discontinue ICI, hospitalization, colonoscopy, steroid therapy, if frustrated, Infliximab 5 mg/kg bw
Grade 4	Stop ICI, hospitalization, steroid therapy, or infliximab

Abbreviations: CT CAE—Common Terminology Criteria for Adverse Events; ICI—immune checkpoint inhibitors.

**Table 7 cancers-16-01371-t007:** General recommendations for the management of immunotherapy-associated side effects.

CTC AE (irAE)	Actions
I	Continue ICI with close monitoring
II	Discontinue therapy until improvement to grade 1, consider corticosteroids if necessary (in case of hypothyroidism: continue ICI, start with L-thyroxine under close clinical and laboratory control of TSH, T3, T4)
III	Discontinue ICI Administration of corticosteroids Infliximab, if no improvement under corticosteroids within 48–72 h Consider biopsy Multidisciplinary management
IV	Termination of therapy (exception: endocrinopathies with improvement through hormone substitution)

## Data Availability

Data are contained within the article.

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
