# Peer review of "Occurrence and Management of Immunotherapy-Associated Adverse Events in Patients with Gynecological Cancers"

_cancers, 2024, doi:10.3390/cancers16071371_

Round 1

Reviewer 1 Report

Comments and Suggestions for Authors

The authors describe the occurence of adverse events in 61 patinets with gynecological malignancies under immunotherapy. As mentioned by the authors in the discussion the restrospective study is limited by the relatively low number of patients. However, immunotherapy is more and more used in patients with different tumor entities and therapy-associated morbidity plays a pivotal role for considering individual treatment regimens. Thus, the manuscript presents a valuable contribution to the field of immunotherapy in patients with gynecological malignancies.

Comments:
#1) As the timing of complications is addressed in the introduction the development of late complications and especially complications after the end of treatment might also be discussed in the revised manuscript.

#2) Figure 2 should be deleted. The number and percentage of overall complications is metioned within the menuscript and does not need additional visualization.

#3) Two patients with ECOG Performance Status 4 have been treated within the described cohort. The indication for systemic treatment of these moribund patients might be added to the revised manuscript.

#4) As mentioned by the authors the duration of ICI therapy was an independent factor for the development of irAEs. Is there a Cutt-off value or definition of 'duration'? This point should be discussed in the revised manuscript.

Author Response

Point-by-point response to the reviewers’ comments

We want to thank the reviewer for taking the time to review this manuscript. We appreciate the reviewers’ comments, and we have prepared a revised version of the manuscript. Modifications in the manuscript are highlighted in red. Please find enclosed our detailed responses to the reviewers’ comments and suggestions:

Reviewer 1:

“The authors describe the occurrence of adverse events in 61 patients with gynecological malignancies under immunotherapy. As mentioned by the authors in the discussion the retrospective study is limited by the relatively low number of patients. However, immunotherapy is more and more used in patients with different tumor entities and therapy-associated morbidity plays a pivotal role for considering individual treatment regimens. Thus, the manuscript presents a valuable contribution to the field of immunotherapy in patients with gynecological malignancies.”

Suggestions:

Comment 1: As the timing of complications is addressed in the introduction the development of late complications and especially complications after the end of treatment might also be discussed in the revised manuscript.

Response 1: Many thanks for this useful and important comment. Since immunotherapy-associated side effects can be closely related to the application of immunotherapy but can also occur months and even years after the end of therapy, it is essential to inform patients and their physicians about the timing of immunotherapy-associated side effects. We added the following paragraph in the revised version of the manuscript (line 93-101, page 3): “A review article by Martins et al. graphically compares the kinetics of the most important immunotherapy-related adverse events during immunotherapy with anti-PD(L)1 antibodies vs. anti-CTLA-4 antibody therapy and illustrates that immunotherapy-related adverse events differ significantly in their kinetics [17]. While skin reactions usually occur in close time relation to the immunotherapy, endocrinopathies can also occur with a time delay or persist even after the end of therapy. The kinetics of immunotherapy-associated side effects depend not only on the organ affected or the type of side effect, but also on the type of immune checkpoint inhibitor used, with clear differences between anti-CTLA-4 antibodies and anti-PD-(L)1 antibodies.”

Comment 2: Figure 2 should be deleted. The number and percentage of overall complications is mentioned within the manuscript and does not need additional visualization.

Response 2: Thank you for your feedback. Following your recommendation, we removed Figure 2 from the manuscript. The total number and percentage of complications are indeed already detailed in the text and Table 2 and do not necessitate additional visualization. We agree that streamlining the presentation will enhance the clarity of our findings.

Comment 3: Two patients with ECOG Performance Status 4 have been treated within the described cohort. The indication for systemic treatment of these moribund patients might be added to the revised manuscript.

Response 3: Thank you for this valuable comment. The two patients classified as ECOG 4 were diagnosed with metastatic breast cancer or metastatic endometrial cancer and had received extensive prior treatment. In both cases, the decision to start immunotherapy was made using shared decision-making principles, taking into account the patient's strong therapeutic preferences. In addition, alternative options, including best supportive care, were discussed in detail. Throughout the course of therapy, these patients were closely monitored to assess tolerability, with the aim of optimizing their quality of life in the setting of their challenging clinical circumstances.

We added the following paragraph in the revised version of the manuscript (line 166-170, page 5):

“However, two patients were classified as ECOG 4. These patients were diagnosed with metastatic breast cancer or metastatic endometrial cancer and had received extensive prior treatment. In both cases, the decision to start immunotherapy was made using shared decision-making principles, taking into account the patient's strong therapeutic preferences.”

Comment 4: As mentioned by the authors the duration of ICI therapy was an independent factor for the development of irAEs. Is there a Cutt-off value or definition of 'duration'? This point should be discussed in the revised manuscript.

Response 4: Thank you for your helpful comment. We have added the following paragraph to the revised version of the manuscript to make the point more clearly understandable (line 203-206, page 8):

“The duration of immunotherapy was defined as the time from initiation to completion/discontinuation of therapy (due to tumor progression or toxicity) and was a median of 32 weeks. It varied from 4 months as the shortest duration of therapy to 48 weeks as the longest duration of therapy.”

Reviewer 2 Report

Comments and Suggestions for Authors

overall a decent review article. 

Please clean up figure 2. 

The authors present the topic Occurrence and Management of Immunotherapy-Associated: Adverse Events in Patients with Gynecological Cancers as a review 

The authors present data from their own hospital which has a diverse set of patients why gynecological disease.

One of my concerns is comparing the use of ICIs across these tumor types may mask the cell contextual nature of the immune therapy.  

Furthermore the sample size in each gynecological groups seems quite small for a clinical study, which limits statistical robustness. 

May I ask if we know whether these patients presented with any of the adverse events such as hypothyroidism before being treated with PDL-1? Or how would we know?

Also a general comment regarding manuscript flow. It seems some of the data is simply being reported on the cohort of individuals in theit hospital without a clear hypothesis?

Please elaborate these points further, as this manuscript may not belong in Cancers. 

Comments on the Quality of English Language

n/a

Author Response

Point-by-point response to the reviewers’ comments

We want to thank the reviewer for taking the time to review this manuscript. We appreciate the reviewers’ comments, and we have prepared a revised version of the manuscript. Please find enclosed our detailed responses to the reviewers’ comments and suggestions:

Reviewer 2

„overall a decent review article.“

Comment 1: Please clean up figure 2.

Response 1: Thank you for your feedback. Following your recommendation, we removed Figure 2 from the manuscript. The total number and percentage of complications are indeed already detailed in the text and Table 2 and do not necessitate additional visualization. We agree that streamlining the presentation will enhance the clarity of our findings.

Comment 2: The authors present the topic Occurrence and Management of Immunotherapy-Associated: Adverse Events in Patients with Gynecological Cancers as a review. The authors present data from their own hospital which has a diverse set of patients why gynecological disease.

One of my concerns is comparing the use of ICIs across these tumor types may mask the cell contextual nature of the immune therapy. 

Response 2: Thank you very much for this important comment. Indeed, we acknowledge the heterogeneity of our patient cohort in terms of tumor entities, which is mentioned in the discussion section as one of the limitations of the study. However, it's important to recognize that while the contextual nature of immunotherapy may vary across tumor types, the basic mechanism of action of immune checkpoint inhibitors and their associated side effects remain relatively consistent. This consistency is reflected in the recommendations of international oncology societies (such as ASCO or ESMO) for the management of immunotherapy-related adverse events, which generally do not differentiate between individual entities [1, 2]. Nevertheless, it's important to recognize the significant differences at the (immune) cellular level between different tumor entities, which may be due to differences in immunogenic potential based on mutational load [3]. Despite these differences, our study focuses primarily on analyzing the incidence, diagnosis, and management of immunotherapy-related adverse events in a cohort of 61 patients with gynecologic malignancies receiving anti-PD(L)-1 antibodies.

Comment 3: Furthermore the sample size in each gynecological groups seems quite small for a clinical study, which limits statistical robustness.

Response 3: Thank you, that is a very valid comment. The small number of cases has been pointed out and taken into account as a limiting factor in the discussion of our results. Despite the increasing use of immune checkpoint inhibitors in gynecologic oncology, the relatively small sample size underscores the novelty of this therapeutic approach and highlights the importance of understanding and managing immunotherapy-related adverse events. Our findings underscore the need for tailored management strategies to optimize the safety and efficacy of immunotherapy in cancer patients.

Comment 4: May I ask if we know whether these patients presented with any of the adverse events such as hypothyroidism before being treated with PDL-1? Or how would we know?

Response 4: Thank you for your question. This information would be obtained by taking the patient's medical history and by laboratory chemistry tests prior to starting immune checkpoint inhibitor therapy.

Comment 5: Also a general comment regarding manuscript flow. It seems some of the data is simply being reported on the cohort of individuals in their hospital without a clear hypothesis?

Response 5: We appreciate your feedback regarding the manuscript's flow. Our paper, submitted as an invited contribution to the Special Issue on "How to Properly Diagnose and Treat Immune-Related Adverse Events during Immunotherapy in Patients with Cancer: Discussion between Specialists," focuses on analyzing the occurrence, diagnosis, and management of immunotherapy-associated adverse events in patients with gynecologic malignancies. The data are presented and discussed within the context of existing literature, aiming to provide valuable insights despite the acknowledged limitations.

Comment 6: Please elaborate these points further, as this manuscript may not belong in Cancers.

Response 6: We have attempted to address the points raised in your comments to provide further clarification and context regarding the scope and relevance of our manuscript in the field of oncology.

  1. Haanen, J., et al., Management of toxicities from immunotherapy: ESMO Clinical Practice Guideline for diagnosis, treatment and follow-up. Ann Oncol, 2022. 33(12): p. 1217-1238.
  2. Brahmer, J.R., et al., Management of Immune-Related Adverse Events in Patients Treated With Immune Checkpoint Inhibitor Therapy: American Society of Clinical Oncology Clinical Practice Guideline. J Clin Oncol, 2018. 36(17): p. 1714-1768.
  3. Schumacher, T.N. and R.D. Schreiber, Neoantigens in cancer immunotherapy. Science, 2015. 348(6230): p. 69-74.

Reviewer 3 Report

Comments and Suggestions for Authors

The authors have written a report on the occurrence and management of adverse events after administering immunotherapy in patients with gynecological cancers. I have few suggestions listed below. Overall the manuscript is well written and illustrated. 

1. The introduction can be expanded to include some history of immunotherapy; specifically authors can include when clinicians started using immunotherapy as a mode of treatment, other cancers treated with immunotherapy, their success levels, caveats etc. The paragraph on irAEs is very good. 

2.  The authors have to provide a rationale for why they decided to study gynecological tumors and not other tumors. 

3. How long should an adverse event occur before it is counted as an AE? How do authors know if it is related to the immunotherapy or not?

4. The grades of irAE can be explained further in the introduction. 

5. Have patients grade of AE increased or decreased over time during treatment of malignancy? Or has the AE grade remained constant?

Author Response

Point-by-point response to the reviewers’ comments

We want to thank the reviewer for taking the time to review this manuscript. We appreciate the reviewers’ comments, and we have prepared a revised version of the manuscript.  Modifications in the manuscript are highlighted in red. Please find enclosed our responses to the reviewers’ comments and suggestions:

Reviewer 3

“The authors have written a report on the occurrence and management of adverse events after administering immunotherapy in patients with gynecological cancers. I have few suggestions listed below. Overall the manuscript is well written and illustrated.”

Comment 1: The introduction can be expanded to include some history of immunotherapy; specifically authors can include when clinicians started using immunotherapy as a mode of treatment, other cancers treated with immunotherapy, their success levels, caveats etc. The paragraph on irAEs is very good.

Response 1: We appreciate very much your comment on our introduction and have already added a few sentences regarding the history and development of immunotherapy. The new paragraph reads as follows (line 45-55, page 2): 

“Since 1957, when Thomas and Burnett suggested that tumor cells could induce an immune response, a number of immunotherapeutic strategies have been approved by the Food and Drug Administration (FDA). Cytotoxic T-lymphocyte-associated Protein 4 (CTLA-4) was the first major immunological checkpoint. Preclinical experiments in 1996 showed that CTLA-4 blockade with monoclonal antibody therapy resulted in tumor re-gression in mice [2]. In 2011, the FDA approved the first antibody to block CTLA-4, ipili-mumab, for the treatment of advanced melanoma. In the decade following this approval, several ICIs that block other immunological checkpoint pathways such as programmed cell death 1 (PD-1) and its ligand, PD-L1, have been approved by the FDA. Although mel-anoma is considered as one of the most immunogenic malignancies, the therapeutic indi-cations for ICIs are increasing in other tumors [3].”

Comment 2:  The authors have to provide a rationale for why they decided to study gynecological tumors and not other tumors.

Response 2: Thank you very much for your notice regarding our study population. The main reason for choosing these tumors is because we only treat patients with gynecological malignancies in the division of gynecologic oncology.

Comment 3: How long should an adverse event occur before it is counted as an AE? How do authors know if it is related to the immunotherapy or not?

Response 3: Thank you very much for your question.

Immunotherapy-related adverse events (IrAEs) usually occur in close temporal relation to the use of immune checkpoint inhibitors but can also occur months and years after completion of therapy. IrAEs can occur at any time during treatment, up to months after discontinuation. However, the occurrence of irAEs also depends on the type of ICi used (earlier with anti-CTLA-4-containing regimens compared to PD(L)-1) and the type of irAE (cutaneous and gastrointestinal irAEs occur earlier than endocrine and renal irAEs). Delayed or late-onset irAEs are defined as occurring 3 months or later after discontinuation of ICI. In contrast to conventional chemotherapy, boosting the immune system results in a constellation of inflammatory toxicities (irAEs). The irAEs described in our report are identified as immunotherapy-related adverse events, taking into account their specific clinical presentation. However, in many cases the diagnosis was confirmed by other specialists involved in the diagnosis and treatment of our patients. In order to specifically address the kinetics of immunotherapy-associated side effects, we have included the following paragraph in the revised version of the manuscript (line 93-101, page 3):

“A review article by Martins et al. graphically compares the kinetics of the most important immunotherapy-related adverse events during immunotherapy with anti-PD(L)1 antibodies vs. anti-CTLA-4 antibody therapy and illustrates that immunotherapy-related adverse events differ significantly in their kinetics [17]. While skin reactions usually occur in close time relation to the immunotherapy, endocrinopathies can also occur with a time delay or persist even after the end of therapy. The kinetics of immunotherapy-associated side effects depend not only on the organ affected or the type of side effect, but also on the type of immune checkpoint inhibitor used, with clear differences between anti-CTLA-4 antibodies and anti-PD-(L)1 antibodies.”

Comment 4:  The grades of irAE can be explained further in the introduction.

Response 4: We appreciate your comment and have therefore added the explanation of the grades of irAE in the introduction (see line 105-108).

“These recommendations also grade the severity of adverse events according to the Com-mon Terminology Criteria for Adverse Events (CTCAE) version 5.0. on a scale from 1 to 5 (1 = mild, 2 = moderate, 3 = severe, 4 = life threatening, and 5 = death related to toxicity) and are therefore an important tool in daily clinical practice.”

Comment 5: Have patients grade of AE increased or decreased over time during treatment of malignancy? Or has the AE grade remained constant?

Response 5: The grade of irAE was first noticed at the onset of irAE during treatment of malignancy. After the beginning of the specific treatment at the onset of irAE, the grade of irAE has decreased.

Round 2

Reviewer 1 Report

Comments and Suggestions for Authors

The revised amunscript has significantly improved and is suitable for publication in Cancers in its present from.